# Prothymosin α Plays Role as a Brain Guardian through Ecto-F_1_ ATPase-P2Y_12_ Complex and TLR4/MD2

**DOI:** 10.3390/cells12030496

**Published:** 2023-02-02

**Authors:** Hiroshi Ueda

**Affiliations:** 1Department of Pharmacology and Therapeutic Innovation, Graduate School of Biomedical Sciences, Nagasaki University, Nagasaki 852-8521, Japan; ueda1hiroshi@icloud.com; 2Graduate Institute of Pharmacology, National Defense Medical Center, Neihu, Taipei 114201, Taiwan

**Keywords:** cell-death-mode switch, ecto-F_1_ ATPase, G_i_ coupling receptor, TLR4, MMPs, neurogenesis

## Abstract

Prothymosin alpha (ProTα) was discovered to be a necrosis inhibitor from the conditioned medium of a primary culture of rat cortical neurons under starved conditions. This protein carries out a neuronal cell-death-mode switch from necrosis to apoptosis, which is, in turn, suppressed by a variety of neurotrophic factors (NTFs). This type of NTF-assisted survival action of ProTα is reproduced in cerebral and retinal ischemia–reperfusion models. Further studies that used a retinal ischemia–reperfusion model revealed that ProTα protects retinal cells via ecto-F_1_ ATPase coupled with the G_i_-coupled P2Y_12_ receptor and Toll-like receptor 4 (TLR4)/MD2 coupled with a Toll–IL-1 receptor domain-containing adaptor inducing IFN-β (TRIF). In cerebral ischemia–reperfusion models, ProTα has additional survival mechanisms via an inhibition of matrix metalloproteases in microglia and vascular endothelial cells. Heterozygous or conditional ProTα knockout mice show phenotypes of anxiety, memory learning impairment, and a loss of neurogenesis. There are many reports that ProTα has multiple intracellular functions for cell survival and proliferation through a variety of protein–protein interactions. Overall, it is suggested that ProTα plays a key role as a brain guardian against ischemia stress through a cell-death-mode switch assisted by NTFs and a role of neurogenesis.

## 1. Introduction

Stroke is one of the most serious brain diseases in terms of high mortality rate due to a reduced blood flow (https://www.who.int/data/global-health-estimates (accessed on 9 December 2020)). Stroke is caused by vascular-obstruction-related cerebral ischemia–reperfusion (in approximately 80% of cases) and vascular-rupture/bleeding-related hemorrhage. In the ischemic core, neurons die due to necrosis caused by energy crisis and release cytotoxic substances, followed by neuroinflammation and cell-death expansion. In the penumbra, on the other hand, apoptotic cell death is often found, but it does not cause cell-death expansion since fragmented cells by apoptosis are phagocytosed by microglia in the vicinity of the brain [1]. Unlike uncontrollable necrosis, apoptosis is characterized as programmed cell death that occurs through multiple molecular events. Current studies propose the view that necroptosis plays roles in the permeabilization of plasma membranes and lysosomes, leading to a necrotic expansion [2]. The necrosome, which comprises receptor-interacting serine/threonine protein kinase 1 and 3 (RIPK1 and RIPK3), plays key roles in this necroptosis [3,4]. However, it is still unclear whether programmed necroptosis inhibitors could totally block cerebral ischemia-induced necrosis. Thus, many studies have attempted to develop apoptosis inhibitors against the neuronal death in stroke. However, thus far, these trials have been unsuccessful, possibly because dead cells, killed by apoptosis, are phagocytosed by microglia in the brain [2] and may terminate cell-death expansion.

## 2. Discovery of Necrosis-Inhibiting Factors

When searching for survival factors from the primary culture of freshly prepared cortical neurons from the embryonic cortex of a 17-day-old rat, we found that large amounts of neurons died due to necrosis within 12 h in low-density (LD; 1 × 10^5^ cells/cm^2^) conditions, while in high-density (HD; 5 × 10^5^ cells/cm^2^) conditions, 80% of neurons survived at 12 h in the absence of serum or any supplementary proteins [5]. For the primary culture, the cortex was dissected from the brain in the L-15 medium by carefully removing other adjacent brain regions, such as the septum, hippocampus and striatum. The cortex was then cut into pieces and incubated in Ca^2+^, Mg^2+^-free PBS, containing trypsin and DNase I, followed by trituration and resuspension in a serum-free DME/F-15 medium without any supplements. Freshly prepared cells, which were cultured at 37 °C in a humidified 5% CO_2_-95% air atmosphere, contained more than 97% neurofilament-positive, while less than 3% contained glial fibrillary acidic protein (GFAP)-positive cells. After conducting transmission electron microscopy, neurons in the LD culture showed a lack of mitochondria swelling, a loss of electron density in cytosol, and plasma membrane destruction, while neurons in the HD culture showed nuclear fragmentation, but mitochondria appeared healthy, and the electron density was normal. In addition, former cells in the LD culture showed intense propidium iodide (PI) in the nucleus, while latter cells in the HD culture showed intense immunocytochemical signals for activated caspase 3. However, all neurons in the HD culture died at 48 h. The addition of the CM from the HD culture to the LD culture also changed the cell death mode from the necrosis showing PI staining to apoptosis, which was evaluated via the immunostaining of annexin V (3 h), activated caspase 3 (12 h) and TUNEL (24 h). It should be noted that very weak PI signals were also observed in CM-treated cells at 24 h, and complete cell death was observed at 48 h, suggesting that so-called necroptosis [6,7] may also occur in a small population of dead cells at a later time point. Since CM is expected to contain necrosis-inhibitory factors, we chose to isolate the factors by measuring the ability to suppress PI signals in the LD culture. After several steps of chromatography, a MALDI-TOF-MS/MS analysis, as the final step, revealed the presence of prothymosin alpha (ProTα), a nuclear acidic protein made from 111 amino acids, starting with acetylated serine [5]. ProTα is distributed everywhere throughout whole-body organs and various brain regions [8]. In the brain, ProTα is found in neurons, astrocytes, and microglia. It is localized in the nucleus of neurons and astrocytes [9], while it is not localized in the nucleus of microglia (unpublished data). Furthermore, ProTα plays multifunctional roles of cell robustness in genomic, epigenetic, and nongenomic mechanisms [10].

## 3. Cell-Death-Mode Switch by ProTα [5]

### 3.1. Mechanisms Underlying ProTα-Induced Inhibition of Neuronal Necrosis

Recent reports described the nature of programmed necrosis and necroptosis [6,7], although unregulated necrosis was first discovered several years ago [11]. The idea that the energy crisis or deficiency of cellular ATP plays key roles in the mechanisms of necrosis is still valid. In our model of neuronal necrosis, the cellular contents of ATP rapidly decreased to 25% 6 h after the start of the LD culture, and the influx of ^3^H-2-D-glucose (2-DG) decreased to 10% at 2 h. A similar rapid decrease in ATP levels (20% at 6 h) was observed under in vitro ischemia conditions, using a low-oxygen (<0.4% O_2_) and low-glucose (1 mM) solution/LOG, followed by a replacement with a DME/F-15 medium containing 5% horse serum and 5% FBS under 5% CO_2_ conditions. These findings are in accordance with the fact that glucose transporters 1 and 4 (GLUT1 and 4) are endocytosed under the same in vitro ischemia–reperfusion stress conditions. The addition of ProTα (80 nM) to the neuronal culture under ischemia–reperfusion conditions externalized GLUT1/4 and reversed cellular ATP levels to 75%. The levels of cell surface GLUT1/4 were evaluated using Western blot analysis after the biotinylation of cell-surface proteins, followed by pull-down using streptavidin-conjugated beads. ProTα reversed the LOG-reperfusion-induced endocytosis of GLUT1/4, and this effect was inhibited by treatments with pertussis toxin as an inhibitor of receptor coupling to G_i/o_, phospholipase C (PLC) inhibitor, and antisense oligodeoxynucleotide (AS-ODN) for protein kinase C (PKC) β_2_. Although G_q/11_-mediated PLC activation is widely accepted, G_i_-mediated mechanisms are also reported in several preparations [12]. We first demonstrated G_i_-PLC coupling through a reconstitution experiment, in which brain membrane G_i_-coupled receptor-mediated PLC activation (Ins1,4,5-P3 production or Ca^2+^ mobilization), abolished by pertussis toxin treatment, was reversed by a reconstitution with purified G_i1_, but not G_o_ [13,14,15]. From these findings, we propose the hypothesis that ProTα inhibits necrosis via a recovery of glucose influx and cellular ATP levels, and underlying signal transduction mechanisms are mediated by an externalization of GLUT1/4 through the activation of putative G_i_-coupled ProTα receptor, PLC, and PKCβ_2_ (Figure 1).

### 3.2. Mechanisms Underlying ProTα-Induced Apoptosis-Induction

Interestingly, ProTα caused typical apoptotic features at 12 h after the addition to the LD culture. To understand its underlying mechanisms, we first studied the activation of various caspases. ProTα activated caspase-3 and caspase-9 [1,16], but not caspase-8 or caspase-12, suggesting that the mitochondria–apoptosome pathway, but not the death receptor [17] or endoplasmic reticulum stress pathways [18], is mainly involved in the apoptosis induction. This view is supported by the following findings. ProTα upregulated proapoptotic Bcl-2 family proteins, Bax and Bim, while downregulating the antiapoptotic Bcl-2 and Bcl-xL. The Bax upregulation was suppressed by the antisense oligodeoxynucleotide (AS-ODN) against PKCβ_1_ or PKCβ_2_, but not PKCα. In accordance with the findings that upregulated Bax opens a mitochondrial permeability transition pore (PTP) channel [19], ProTα induced cytochrome c (Cyto.c) release from mitochondria. Cyto.c release was suppressed by Bax inhibitor peptide V5 (BIP-V5), which is reported to block the translocation of Bax to mitochondria [20]. ProTα activated caspase-9, possibly through an apoptosome formation induced by released Cyto.c [21], followed by caspase-3 activation. As often reported, apoptosis may be driven through caspase-3/CAD (caspase-activated end-DNase) activation [22]. Indeed, in the late phase (24 and 48 h) after ProTα addition, the time-dependent signal of TUNEL was observed (terminal deoxynucleotidyl transferase-mediated dUTP nick end labeled). This reflects nuclear fragmentation, which is a typical feature of apoptosis. The TUNEL signal was completely inhibited by zVAD-fmk, a pan caspase inhibitor [23]. Interestingly, activated caspase-3 degrades poly (ADP-ribose) polymerase (PARP), which consumes abundant ATP for the restoration of damaged DNA [24]. In other words, the caspase-induced degradation of PARP may delay the occurrence of necrosis by reducing ATP consumption. This delay of cell death appears to be important for cell survival in vivo during the approach of anti-apoptotic NTFs existing in the brain or when induced by ischemia stress. We observed that apoptosis converted by ProTα in the LOG stress model was completely abolished by NTFs, such as the nerve growth factor, brain-derived neurotrophic factor (BDNF), basic fibroblast growth factor, or interleukin-6, which have no effects on the necrosis of cells without ProTα [5]. All of these findings are summarized in Figure 2. It should be noted that necrosis manifested by PI-staining reappeared 48 h after ProTα addition in the presence of zVAD-fmk. Although this type of necrosis may be known as necroptosis [25], it is not clear whether caspase inhibition occurs in vivo.

## 4. In Vivo Beneficial Actions of ProTα

### 4.1. Cell-Death-Mode-Switch in Retinal Ischemia–Reperfusion Model

An ischemia–reperfusion model of the retina is useful for studying the cellular and molecular mechanisms of neuronal cell death. In a mouse model, retinal ischemia was induced by a hydrostatic intraocular pressure of 130 mmHg for 45 min [26]. This method allows pharmacological actions to be assessed via the intravitreous injection (i.vt.) of small amounts of compounds in terms of morphology and electrophysiological functions. As cellular layers are clearly separated, morphological changes in each cell layer can be assessed by immunohistochemistry, as well as hematoxylin and eosin (H&E) staining. Using the retinal ischemia–reperfusion model, the thickness of the retina was reduced to approximately 50% over time, but the damage became maximal at day 7. Functional changes were assessed by electroretinography (ERG), in which, at first, a negative current amplitude (a wave) reflected the activity of photoreceptor layers, while the next positive current amplitude (b wave) carried out the activities of the ganglion cell layer and inner nuclear layers, including bipolar, amacrine cell layers, etc. Retinal ischemia–reperfusion stress damaged all these cell layer activities with some minor exceptions, suggesting that hydrostatic pressure causes damage in whole retinal cell layers. The intravitreous injection (i.vt.) of ProTα at 1 pmol/eye given 24 h after the reperfusion completely reversed the reduction in retinal thickness in the ganglion cell layer (GCL), inner nuclear cell layer (INL), and outer nuclear cell layer (ONL) at day 7. Similar results were also observed when ProTα was administered 30 min before or 3 h after reperfusion. The functional damage in the electroretinogram (ERG) was also reversed by ProTα in a dose-dependent manner in the range from 0.01 to 1 pmol/per eye (i.vt.). A complete recovery was observed with 1 pmol (i.vt.) and 100 μg/kg (i.v.). The advantages of ProTα in this model were dose-related and significant when it was administered (i.vt. or i.v.) 30 min before, 3 h after, or 24 h after reperfusion, and the best time point for injection was 3 h after the reperfusion in both cases.

ProTα blocked both necrosis (PI staining), with a peak effect at day 1, and apoptosis (caspase 3-like activity), with a peak effect at day 3. Interestingly, ProTα upregulated BDNF and erythropoietin (EPO) in the retina, while the treatment with anti-BDNF or anti-EPO IgG reversed the occurrence of apoptosis, but not necrosis [26]. Thus, it is evident that ProTα converts necrosis to apoptosis, which is then inhibited by NTFs. The guardian roles of ProTα under retinal ischemia–reperfusion conditions were further supported by the findings that the ischemic stress depletes ProTα in the inner nuclear cell layer, and the partial damage at day 4 after the reperfusion was deteriorated by pretreatment with AS-ODN for ProTα or anti-ProTα IgG.

### 4.2. Preconditioned ProTα-Induced Retinal Protection via TLR4/MD2-TRIF Pathway

Some fascinating studies report that lipopolysaccharide (LPS) mediates tolerance to ischemic injury via Toll-like receptor 4 (TLR4) signaling, and ProTα induces the expression of type 1 interferon and suppresses the HIV-1 gene expression in macrophages through TLR4/MD2 [27,28,29]. Biophysical evidence for a direct ProTα-TLR4/MD2 interaction was obtained via gravimetric analyses using recombinant proteins and quartz crystal microbalance/QCM [30]. The K_D_ value of ProTα for the binding to the TLR4/MD2 complex was estimated to be 16.07 μM. On the other hand, a molecular dynamics simulation revealed that LPS binds to the MD2 moiety of this complex, while ProTα binds to both the MD2 and TLR4 moieties [30]. The LPS-mediated tolerance to ischemic injury [29] was reproduced using a retinal ischemia–reperfusion model [31]. In this report, the prevention of ischemia–reperfusion-induced damage was maximal (approximately 50%), with LPS preconditioning (i.vt.) at 48 h, and partial with the preconditioning at 24 h prior to ischemia–reperfusion. This inhibition was abolished in TLR4^−/−^ mice or via pretreatment with TLR4-antibodies. ProTα-preconditioning also prevented retinal ischemic damage, and its effective dose was similar to the case with post-ischemia treatment (maximal at 1 pmol, i.vt.). The beneficial actions were abolished in TLR4^−/−^ mice. Immunohistochemical studies show that the percentage of ProTα-rescued NeuN-positive GCL cells, Chx10-positive bipolar cells, Syntaxin-1-positive amacrine cells, and Hoechst33342-positive photoreceptor cells decreased by 75, 50, 10, and 30%, respectively, but these preventive effects were all abolished in TLR4^−/−^ mice. Interesting findings were observed for the cellular mechanisms. ProTα preconditioning mildly increased the number of Iba-1-positive microglia in GCL, INL and ONL, and peak effects were observed at day 2. It should be noted that ProTα suppressed the ischemia–reperfusion-induced increase in the phosphorylated p38-positive microglia population, and this suppression was reversed by the treatment with neutralized TLR4 antibodies or minocycline. The preconditioning of ProTα had no effects on the ischemia–reperfusion-induced activation of GFAP-positive astrocytes. All of these findings suggest that ProTα activates microglia but suppresses the ischemia–reperfusion-induced activation of cytotoxic microglia through TLR4-mediated mechanisms.

Transcriptional studies supported the preconditioned ProTα-induced alternative activation of microglia in the retina [32]. The quantitative polymerase chain reaction data show that ischemia–reperfusion causes huge increases in the levels of mRNAs in injury factors, such as TNFα, IL-1β, IL-6, COX-2, as well as monocyte chemoattractant protein-1 at 24 h after the induction of ischemia in a vehicle-pretreated retina. Meanwhile, ProTα preconditioning markedly suppressed the ischemia–reperfusion-induced elevation of mRNA levels of these injury factors. On the other hand, ProTα alone upregulated cytoprotective factor genes, such as IL1RN, IFIT1, RANTES, SOCS1, SOCS3, and Ship1. All these findings suggest that ProTα transforms the resting microglia into cytoprotective M2-type one, which suppresses the cytotoxicity of M1-type microglia caused by the ischemic stress (Figure 3).

### 4.3. Possible ProTα Receptor Mechanisms via Ecto-F_1_ ATPase-P2Y_12_ Complex

In a study using a primary culture of cortical neurons, ProTα showed a unique cell-death-mode switch via a putative G_i_-coupled receptor. However, this receptor signaling is not compatible with ProTα preconditioned via a TLR4-TRIF system in the retinal ischemia–reperfusion model. Indeed, the beneficial actions of ProTα given after the reperfusion were not affected in TLR4^−/−^ mice [33]. To identify the G_i_-coupled ProTα receptor, the G_i_-rich lipid rafts fraction of the retinal cell line N18-RE-105 cells was used for affinity cross-linking via Sulfo-SBED, a trifunctional crosslinking reagent with biotin covalently attached to a heterobifunctional reagent [34]. The biotinylated ProTα-binding target proteins were separated by SDS-PAGE. The last step using MALDI-TOF-MS/MS analysis revealed that ProTα binds to ATP5B, a subunit of F_1_ ATPase. In the ELISA assay, the K_D_ values of ProTα to ATP5B, ATP5A1, and combined ATP5A1/5B were 241.33, 67.98, and 55.78 nM, respectively. In the QCM analysis, the interaction between the ProTα and ATP5A1/5B complex was K_D_ = 28.08 nM, lower than the case with the ELISA. Therefore, ProTα binds with a higher affinity to ATP5A1, the alpha subunit of F_1_-ATPase, compared to its beta subunit, ATP5B.

In the retinal ischemia–reperfusion model, on the other hand, pretreatment with a neutralizing antibody against ATP5A1 or ATP5B abolished the retinal protective actions of ProTα with the ERG test [33]. This finding strongly suggests that ecto-F_1_ ATPase plays a key role in the beneficial action of ProTα. In vitro studies revealed that ProTα increased the ATPase activity of the recombinant ATP5A1/5B complex of human ecto-F_1_ ATPase [33]. Therefore, it is fascinating to speculate that the product of ADP may activate P2Y receptors through G_i_ mechanisms, which have been proposed in the cell-death-mode switch of cortical neurons under the conditions of an ischemia–reperfusion model (Figure 1 and Figure 2). Indeed, the pretreatment with antibodies against the cell surface epitope of a G_i_-coupled P2Y_12_ receptor largely reversed the protective actions of ProTα in the retinal ischemia–reperfusion model. Most recently, however, it was reported that G_i_-coupled receptor-mediated PLC activation through a binding of Gβγ is dependent on the additional Gα_q_ binding to PLC [12]. If this Gα_q_ involvement also functions in the case of ProTα mechanisms, ADP may also activate G_q_-coupled P2Y receptors (Figure 4).

ProTα increased the extracellular ATP levels in the culture of HUVEC cells [34], as in the case of thymosin β4 [35]. Since this increase was reversed by pretreatment with neutralizing antibodies against the extracellular epitope of ATP5A1 or ATP5B [34], and ProTα increased ATP hydrolysis activity of recombinant ATP5A1/5B proteins [33], the inhibition of ATP hydrolysis activity of ecto-F_1_ ATPase [36] is unlikely to be involved in this mechanism. However, it remains elusive whether ATPase activation or ATP production plays a role in the ecto-F_1_ ATPase-mediated ProTα actions. Regarding the source of ADP, there are several possibilities. They are the ADP production from ATP by ectonucleoside triphosphate diphosphohydrolase [37], and ecto-F_1_ ATPase [33]. There is an interesting report that adenine nucleotide translocase (ANT) is closely localized with ecto-F_1_ ATPase at the plasma membrane of hepatocytes. This can control the extracellular ADP level depending on the extracellular ADP/ATP ratio, and apolipoprotein A-1 binding to ecto-F_1_ ATPase inhibits ANT and increases the level of extracellular ADP [38]. Such a unique ecto-F_1_ ATPase/P2Y pathway has been identified in the promotion of HDL holoparticle endocytosis via hepatocytes [39] and in the suppression of apolipoprotein A1 in endothelial nitric oxide synthesis and vasorelaxation [40]. In addition, it should be noted that ecto-F_1_ ATPase is also activated by thymosin β4 (Thβ4), which is one of the thymosin 5 peptides, as well as part of ProTα. Freeman et al. [35] reported that Thβ4 binds to and increases extracellular ATP levels via ecto-F_1_ ATPase in HUVEC cells, and they proposed that Thβ4 migrates HUVEC cells through the action of a P2X_4_ receptor. Thus, it is interesting to speculate that ProTα-activated ecto-F_1_ ATPase may use alternative signaling to activate P2X_4_ receptor, which in turn produces BDNF [41].

## 5. ProTα Actions in the Cerebral Ischemia–Reperfusion Model

### 5.1. Inhibition of Cerebral Ischemia-Induced Brain Damage

In the rat model with 1 h MCAO, ProTα at 100 μg/kg (i.p.), administered at 1.5 and 4 h, completely suppressed the brain damage and was evaluated by 2,3,5-triphenyltetrazolium chloride (TTC) staining [42]. ProTα also significantly suppressed MCAO-induced neurological score and mortality. In accordance with the serum-free culture study of cortical neurons, ProTα inhibited necrosis according to PI staining and apoptosis, which were evaluated by activated caspase-3 immunoreactivity. Significantly, the population of apoptotic cell death reappeared when anti-BDNF IgG was given 30 min prior to MCAO, suggesting that ProTα causes a cell-death-mode switch in vivo, and generated apoptosis is suppressed by endogenous BDNF.

On the other hand, when ProTα at 1–100 μg/kg (i.p.) was administered 24 or 48 h after sublethal bilateral common carotid arteries occlusion (BBCAO)-type cerebral ischemia for 30 min, there were dose-dependent reversals of the loss of pyramidal cells at the CA1 hippocampus and memory/learning ability in the step-through-type passive avoidance test on day 28. This study indicates that ProTα has beneficial actions in focal and global ischemia models via systemic administration. The differences in therapeutic time periods for ProTα, provided by a systemic route in either model, may be attributed to the best conditions of blood–brain-barrier permeability.

Interestingly, ProTα has a protective action from MCAO-induced cerebral blood vessel disruption. When biotinylated tomato lectin (1 mg/mL; i.v.) was administered to mice 24 h after the MCAO, followed by staining of brain sections with Alexa Fluor 488 streptavidin, ProTα reversed the reduction in cerebral vascular disruption. ProTα also reversed the IgG and Evans Blue leakage in the brain.

### 5.2. Inhibition of Late tPA-Induced Hemorrhage

In clinical settings, the rapid infusion of tissue plasminogen activator (tPA) is known to prevent brain damage caused by stroke. However, tPA administration is restricted due to its narrow therapeutic time window; late treatment has a risk of causing cerebral hemorrhage [43,44,45,46,47]. In the mouse model, the infusion of tPA (10 mg/kg; i.v.) 4.5 h after the MCAO sporadically caused a cerebral hemorrhage in the ipsilateral cortex and striatum. The hemorrhage was completely abolished by ProTα (100 μg/kg; i.p.) administered with tPA [48]. In this paradigm of ProTα treatment with tPA at 4.5 h, most mice died at day 7. However, ProTα, administered twice at 2 and 4.5 h after the start of MCAO, largely prevented mortality and hemorrhages. Ischemia-induced vascular damage is known to be related to the degradation of tight-junction proteins via the activation or upregulation of matrix metalloproteases (MMPs). In the MCAO model, high levels of immunoreactive MMP-2 were observed in microglia, but when tPA was given 4.5 h after the MCAO, MMP-2 was also observed in CD31-positive vascular cells and microglia. The co-treatment of ProTα with tPA abolished MMP-2 expression in microglia and vascular cells [48]. On the other hand, the protein levels of tight-junction protein occludin and scaffold protein zonula occludens (ZO)-1, were both decreased via treatment with MCAO and tPA, and this decrease was reversed by the co-administration of ProTα with tPA. As mentioned above, we found that ProTα increased ATP levels in a culture of HUVEC cells through ecto-F_1_ ATPase [34]. Therefore, the cerebral endothelial cell is an important site of ProTα in terms of beneficial actions against late tPA-induced hemorrhage. Further studies to clarify the relationships between ecto-F_1_ ATPase, tPA receptors, and MMP activation are essential.

## 6. In Vivo Role of ProTα Using Transgenic Mice [49]

ProTα is expressed in a wide variety of tissues in the body and possesses multi-functional activities. ProTα plays roles in cell-cycle progression, proliferation, and survival [1,50,51]. ProTα also regulates cell-defensive mechanisms against oxidative stress [52]; its overexpression is associated with the development of pulmonary emphysema [53] and various cancer types [54]. The C-terminal sequence of ProTα produces immune responses via the stimulation of lymphocytes and maturation of dendritic cells [55]. As mentioned above, ProTα plays a protective role in the brain at the time of a stroke. However, little remains known about the physiological and pathophysiological roles of ProTα, aside from the findings of studies on neutralizing antibody and antisense strategies [26]. Recent studies revealed the roles of ProTα in the brain using heterozygous [49] and conditional knock-out mice [56].

### 6.1. Gross Behavioral Activities of ProTα^+/−^ Mice

ProTα-knockout mice were generated using C57BL embryonic stem cells via the Cre-mediated deletion of exons 2 and 3 of the ProTα gene. The number of offspring produced from ProTα^+/−^ mating was significantly lower than for the case of ProTα^+/+^ mice. The birth ratio of ProTα^+/+^: ProTα^+/−^: ProTα^−/−^ was approximately 1:2:0. ProTα^+/−^ showed a significant decrease in body weight and body temperature compared to wild-type (WT) mice. The righting, whisker touch, and ear twitch reflexes were normal in ProTα^+/−^ mice, and there was no significant change in grip strength, latency when falling from an inverted wire mesh, thermal nociception in the hot plate test, and forced motor activity in the accelerating rotarod test. Thus, it is evident that the administration of ProTα^−/−^ in mice should be lethal; however, ProTα^+/−^ mice retained relatively normal physical activities.

### 6.2. Enhanced Anxiety-like Behaviors of ProTα^+/−^ Mice

In the open-field test, ProTα^+/−^ mice showed a significant decrease in total distance travelled, time spent in the center area, vertical activity, and stereotypic counts compared to WT mice, indicating that ProTα^+/−^ mice have decreased locomotor activity. In the anxiety-related light/dark transition test, ProTα^+/−^ mice showed a significant decrease in distance travelled in the light compartment and time spent in the light compartment compared to WT mice. The novelty-induced hypophagia test, measured via the latency of drinking sweetened milk solution and total milk consumption, was used to evaluate environment-induced anxiety-like behavior. In this test, ProTα^+/−^ mice showed a significant tendency to prolong the initial latency compared to latencies found in each home cage group. ProTα^+/−^ mice showed a longer latency to start drinking milk and a significant decrease in total milk consumption, compared to WT mice. In the elevated plus-maze test, however, ProTα^+/−^ mice did not show significant changes in the number of entries into open arms, percentage of entries into open arms, total distance travelled, and percentage of time spent in the open arms. In the marble burying test used to evaluate obsessive compulsive disorder [57,58], ProTα^+/−^ mice also did not show any changes in the number of marbles buried and total distance traveled. All these findings suggest that ProTα^+/−^ mice exhibit hypolocomotor activity and some anxiety-like behaviors.

### 6.3. Impaired Learning and Memory in ProTα^+/−^ Mice

In the cued fear conditioning memory test, mice were given foot shocks with an auditory-conditioned stimulus. When placed in different triangular chambers 2 days after the conditioning, ProTα^+/−^ mice displayed a higher freezing ratio in first 3 min without tone stimulus compared to WT mice. In the KUROBOX test, which involved a stress-free positive cue task, the trajectory of the region of interest (ROI) was evaluated to assess spatial learning and memory [59]. The correct ratio (visits to the station with food compared with visits to all stations) for every 1 h in ProTα^+/−^ mice was significantly lower than that in WT mice, suggesting an impaired spatial discrimination and poor exploratory behavior when searching for food. In the step-through passive avoidance test, on the other hand, ProTα^+/−^ mice showed a significant increase in latency when entering a dark compartment throughout the conditioning session. This behavior may be partly related to an anxiety-prone nature. However, 24 h after the conditioning, the step-through latency in ProTα^+/−^ mice was significantly decreased in the retention trials, suggesting impaired learning and memory.

### 6.4. Declined LTP Induction in ProTα^+/−^ Mice

Planar multielectrode arrays and electronics (MED system) are used worldwide for the study of long-term potentiation (LTP), which involved hippocampal slices associated with memory and learning [60,61,62,63,64]. In this system, each field-excitatory post-synaptic potential (fEPSP) in the Schaffer collateral/CA1 hippocampal pathway was adjusted to evoke a response induced by bipolar constant current pulses (3–50 μA, 0.1 ms). After three theta bursts stimuli, the fEPSP increased for more than 30 min. LTP was significantly suppressed to the level of 50% in ProTα^+/−^ mice.

### 6.5. Decreased Adult Hippocampal Neurogenesis in ProTα^+/−^ Mice

Adult hippocampal neurogenesis plays a role in the maintenance of emotional and cognitive functions [65,66,67]. In BrdU labeling [62], to examine the impairment of adult hippocampal neurogenesis, the number of BrdU-labeled cells in the subgranular zone of ProTα^+/−^ mice was significantly reduced [49]. ProTα^+/−^ mice also had a decreased number of BrdU-positive (Figure 5, *left panels*) or doublecortin-positive cells in the dentate gyrus [49]. When tMCAO was administered for 15 min, BrdU-positive adult neurogenesis increased in the dentate gyrus and was completely abolished in ProTα^+/−^ mice (Figure 5, *right panels*), suggesting that ProTα is involved not only in the naive adult neurogenesis, but also in the stroke-induced increase in neurogenesis (restoration).

### 6.6. Conditional ProTα Knock-Out Mice

Using conditional ProTα knockout mice (Gng7^+cre^-ProTα^−/−^), the product of the cross-breeding of striatum-specific G protein γ subunit gene Gng7^Cre+/+^ mice with ProTα^flox/flox^ mice, brain-region-specific roles of ProTα were investigated [56]. This transgenic mouse demonstrates the deficiency of ProTα in the hippocampus and cerebellum, as well as the striatum in the immunohistochemistry and Western blot analyses. At 10 weeks old, cKO mice experienced a significant deterioration in neurological score and survival ratio when weak tMCAO (15 min) was given, but cKO mice did not show any significant motor dysfunction in the rotarod test in the absence of tMCAO. However, at 20 weeks old, cKO mice showed significant motor dysfunction. A similar age-dependent phenotype was observed in the hypolocomotor activity with less center time in the open-field test. These findings suggest that the environmental social stress from the community for longer periods in the same cage (five mice) may cause brain damage (motor incoordination and anxiety-like behavior) in the absence of ProTα.

## 7. Multiple Intracellular Functions

Nuclear protein ProTα is released from neurons or astrocytes upon ischemia or heat-shock stress [9] and exerts neuroprotective actions through the G_i_-mediated putative ProTα-R complex and TLR4/MD2-TRIF system. In addition to such extracellular functions, ProTα has multiple intracellular functions in cell survival and proliferation through a variety of protein–protein interactions [10]. These include Nrf2-Keap 1 mechanisms, apoptosome formation inhibition, chromatin remodeling epigenetic regulation, and modulation of estrogen receptor activity. Nrf2 (nuclear factor erythroid 2-related factor 2) is a nuclear transcription factor that regulates the expression of several defensive genes (detoxifying enzymes and antioxidant genes) [68,69,70]. In the absence of stress, Nrf2 is trapped by Keap 1 (Kelch-like ECH-associated protein 1), ubiquitinated by the Cul3/Rbx1-dependent E3 ubiquitin ligase, and degraded by the 26S proteasome [71]. As ProTα binds to Keap 1 and releases Nrf2 from the Nrf2-Keap 1 complex [52], it is speculated that ProTα regulates cell-defensive roles of Nrf2. During apoptosis, large amounts of ProTα become located in the cytosol by the caspase-3-mediated cut-off of its C-terminus containing a nuclear localization signal. As (truncated) ProTα inhibits apoptosome formation by interactions with Apaf-1, apoptosis may be auto-regulated. Regarding the nuclear functions, nuclear ProTα epigenetically stimulates gene transcription by binding to histones [72], p300 histone acetyltransferase [73] and CREB-binding protein [74], suggesting that ProTα may play roles in chromatin remodeling [75]. A recent study demonstrated that adeno-associated virus-9-mediated zinc finger E-box binding homeobox2 (ZEB2) delivery, which upregulates ProTα gene expression, stimulated cardiomyocyte-derived angiogenic signals in the injured heart and thereby improved cardiac repair [76]. Further research of the intracellular and extracellular functions of ProTα, which are upregulated through injury-induced ZEB2-mediated mechanisms, is essential.

## 8. Conclusions

In the present study, we propose two types of ProTα receptor mechanisms for playing protective roles against ischemia models. One type is mediated through a putative ProTα receptor complex comprising ecto-F_1_ ATPase coupled to a P2Y_12_ mechanism, and the other type is coupled to a TLR4/MD2-TRIF mechanism. In these mechanisms, ProTα transforms the resting microglia into a cytoprotective M2-type mechanism, which suppresses the cytotoxicity of M1-type microglia caused by ischemic stress. The proposed mechanism is the initial cell-death-mode switch from necrosis to apoptosis, which is subsequently inhibited by NTFs in vivo. Regarding the cerebral ischemia–reperfusion–hemorrhage model, ProTα inhibits the production of MMPs in microglia and vascular endothelial cells. Detailed molecular mechanisms remain elusive, but some of them may be related to the intracellular actions of ProTα. Finally, the study using ProTα^+/−^ mice revealed that the mice have multiple phenotypes, such as decreased LTP, a memory learning deficit, anxiety, and a lack of adult neurogenesis. All these findings suggest that ProTα plays protective roles in the brain (Figure 6).

## Figures and Tables

**Figure 1 cells-12-00496-f001:**
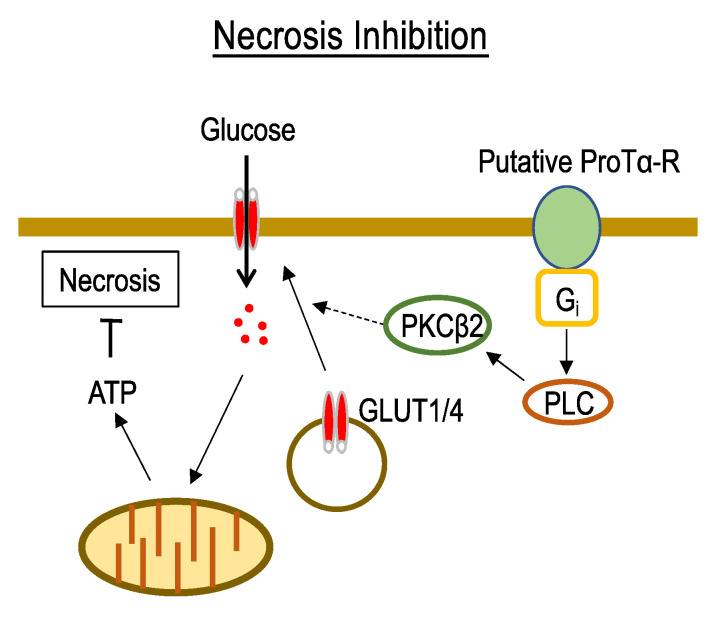
Proposed mechanisms underlying ProTα-induced necrosis inhibition. ProTα activates putative G_i_-coupled receptor, followed by an activation of PLC and PKC-β2, which contributes to the translocation of GLUT1/4 to the plasma membrane. GLUT1/4 causes a glucose influx and increases intracellular glucose levels via the glycolysis pathway. As a result, intracellular ATP levels increase and prevent necrosis. More details are described in the text.

**Figure 2 cells-12-00496-f002:**
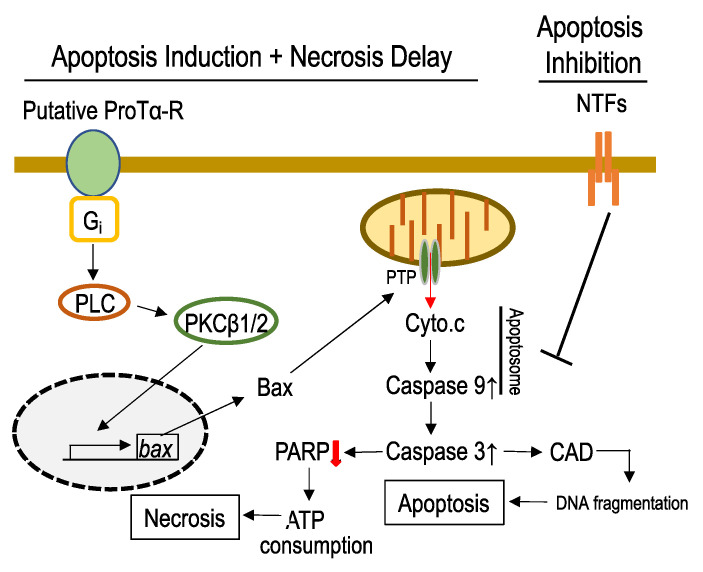
Proposed mechanisms underlying ProTα-induced apoptosis induction. ProTα activates PKC-β1 and β2 through a stimulation of the putative G_i_-coupled receptor and phospholipase C (PLC). PKC-β1 and β2 increase the level of Bax, a proapoptotic Bcl2 family protein, which opens the permeability transition pore (PTP) channel and releases cytochrome c (Cyto.c) from the mitochondria. Released Cyto.c forms the apoptosome, followed by an activation of caspase 9 and caspase 3. Caspase 3 causes apoptosis via DNA cleavage and degrades poly-(ADP-ribose) polymerase (PARP), leading to decreased ATP consumption and the delay of necrosis. While buying time of necrotic cell death, endogenous or ischemia (or ProTα)-induced NTFs, such as BDNF and EPO, suppress apoptotic mechanisms in the brain or retina.

**Figure 3 cells-12-00496-f003:**
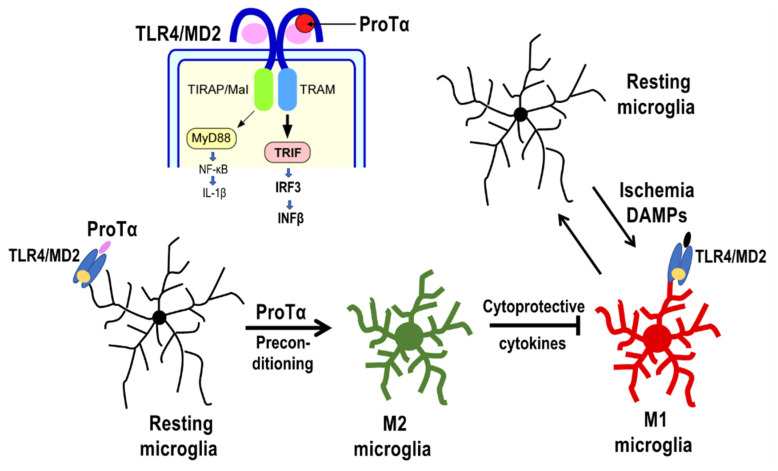
ProTα drives resting microglia into M2 microglia. ProTα drives the TLR4/MD2-mediated Toll–IL-1 receptor domain-containing adaptor inducing IFN-β (TRIF) system. ProTα drives resting microglia into M2 microglia, which show an alternative activation through the production of cytoprotective cytokines and deactivation of M1 microglia, which are induced by ischemia-related damage-associated molecular patterns (DAMPs)/cytotoxic cytokines.

**Figure 4 cells-12-00496-f004:**
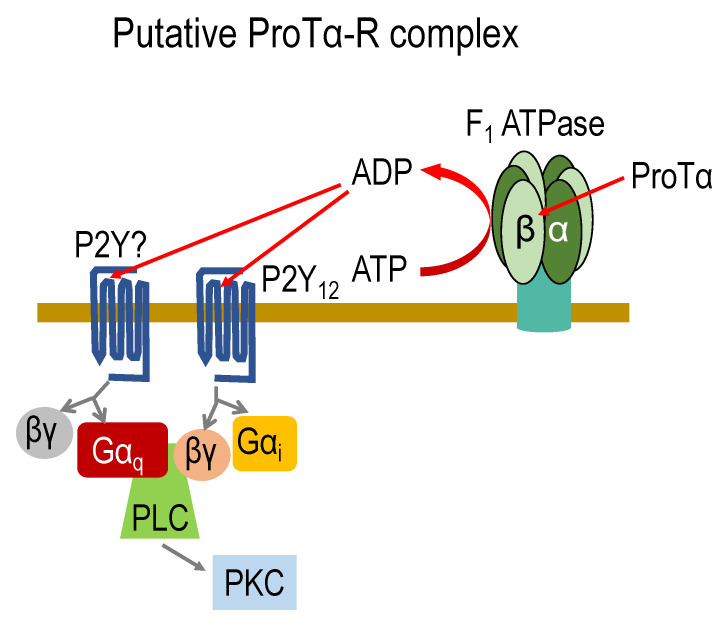
Putative ProTα-receptor complex. ProTα receptor (ProTα-R) complex comprises an ecto-F_1_ ATPase, a G_i_-coupled P2Y_12_ receptor, and an unidentified G_q_-coupled P2Y receptor, based on ProTα increasing the ATPase activity of the recombinant F_1_ ATP5A1/5B complex. In this model, the source of ATP remains elusive. The discussion of an alternative source of ADP in relation to ecto-F_1_ ATPase activity is described in the text.

**Figure 5 cells-12-00496-f005:**
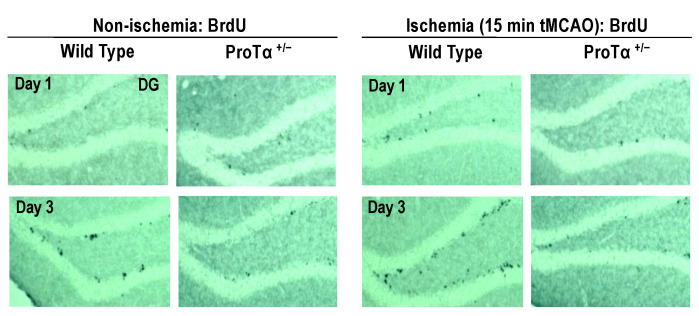
Involvement of ProTα in adult neurogenesis. The results show the representative images showing 5-bromo-2′-deoxyuridine (BrdU)-labeled cells in the sub-granular zone of the hippocampal dentate gyrus (DG) in wild-type and ProTα^+/−^ mice, as reported elsewhere [49]. Mice received intraperitoneal (i.p.) injections of 50 mg/kg BrdU once a day for three consecutive days. All mice were killed 24 h after last BrdU injection. BrdU labeling in the brain was analyzed by immunohistochemistry. Some BrdU signals in the DG of wild-type mice were observed at day 3, but not day 1, showing that it is a de novo signal. In the DG of ProTα^+/−^ mouse, on the other hand, BrdU signals were lost. When transit MCAO (15 min) was given to wild-type mouse, more BrdU signals were observed in the wider region of sub-granular zone of DG, while no BrdU signals were observed in the DG of ProTα^+/−^ mouse. These results suggest that ProTα is involved not only in naturally occurring adult neurogenesis, but also in the stroke-induced increase in neurogenesis.

**Figure 6 cells-12-00496-f006:**
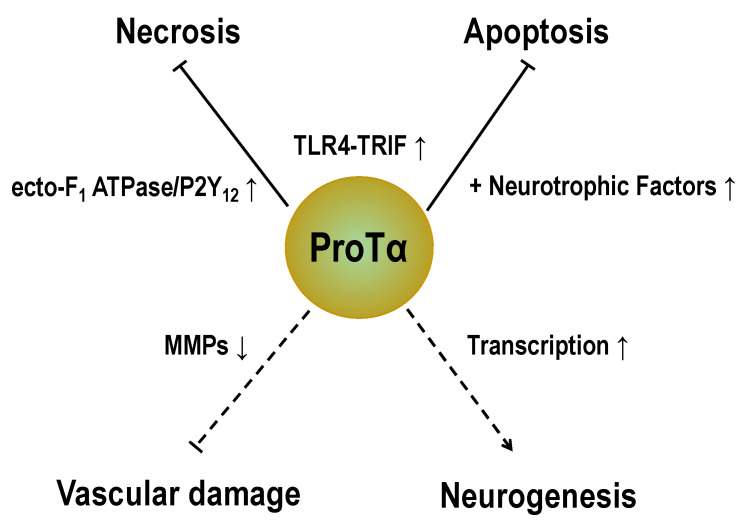
ProTα as a brain-protective molecule. ProTα plays multiple brain-protective roles against stroke. Upon the ischemic stress, nuclear protein ProTα is extracellularly released and suppresses necrotic mechanisms, potentially through the ecto-F_1_ ATPase-P2Y_12_ complex, which is coupled to G_i_-mediated mechanisms. Extracellular ProTα also leads to apoptotic mechanisms via apoptosome-caspase 3 pathways, which delay the occurrence of necrosis. During this period, NTFs, such as BDNF, are produced and inhibit apoptosis. ProTα also inhibits the activation of MMPs and protects the brain from vascular damage. A study using ProTα^+/−^ mice found that ProTα plays key roles in adult neurogenesis under naïve and ischemia conditions. All of these findings suggest that ProTα plays brain-protective roles.

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
