# Peer review of "Prothymosin α Plays Role as a Brain Guardian through Ecto-F1 ATPase-P2Y12 Complex and TLR4/MD2"

_cells, 2023, doi:10.3390/cells12030496_

Round 1
Reviewer 1 Report
In this manuscript, Ueda reviews data on the mode of action of prothymosin alpha in the brain and its impact on neuronal cell death. It is an interesting review, summarizing data on prothymosin alpha’s role as a molecular switch, the mechanisms underlying the observed effects and it opens the field for potential clinical applications in ischemia. Some minor comments that should be considered:
1. Please add a list of abbreviations, as these are too many in text and quite confusing.
2. Needs extensive editing for typos and in some cases also syntax, eg. paragraph 2, line 2, “we found that large amounts”; lines 5-8, needs rephrasing, etc
3. In all figures, omit phrases in bold written over legends, which are a repetition of the first phrase of the figure legends.
4. In Fig. 2, the arrow next to ATP should show decrease not increase.
Author Response
Thank you very much for your kind comments. I revised the manuscript according to the reviewer's comments.
1 Abbreviations were added.
2 English edition by MDPI was done.
3 Figure issues was appropriately corrected.
4 ATP issue in Fig. 2 was appropriately corrected and the related description was added in the text.
Reviewer 2 Report
The authors of the manuscript ‘Prothymosin α plays roles as a brain guardian through ecto-F1 ATPase-P2Y12 complex and TLR4/MD2’ describe the current understanding of the role of Prothymosin α and downstream signaling in protecting against ischemia stress. The subject is interesting and falls well within the scope of the Journal.
In principle, the manuscript is potentially interesting but I have several comments.
Major comments:
1) Sections 2 and 3 both include the detailed method that led to the discovery of ProTα. The first paragraph of section 3 should be merged to section 2, avoiding repetitions.
On the other hand, the nature of ProTα should be better defined in a first paragraph of section 3 including its molecular weight, its expression level in cell types and tissues, its previously described roles and functions, etc.
2) Section 3 and 4:
PLC activation is presented downstream Gi signaling. This is unexpected as PLC/PKC pathways being classically activated downstream Gq (and not Gi). Please comment. Is Gq activation by ProTα has been totally excluded?
3) Section 4.3, page 6:
Kd value does not allow to conclude on the binding mechanisms but on affinity only:
- Please delete: “indicating that Flag-modified ProTα first binds to….azide reaction”.
- Replace “Therefore, ProTα binds to ATP5A1” by “Therefore, ProTα binds with higher affinity to ATP5A1, the alpha subunit of F1-ATPase, compared to its beta subunit, ATP5b”.
- The sentence “ATP5A1-attache ATP5B is biotinylated” is confusing. Please delete it.
4) Section 4.3, page 7:
- “ProTα induced to production of ATP”. Please specify that it is the production of extracellular ATP. It is more likely ProTα increases extracellular ATP level by inhibiting the ATPase activity of F1-ATPase since ATPase activity is the solely reported activity of F1-ATPase at the cell surface (Nature 2003, 421, 75–79; Arter. Thromb Vasc Biol 2009, 29, 1125–1130). Please add this possibility and references..
- “There are 2 posibilities for the ADP production…” Please add “extracellular ADP production”. Extracellular ADP production can be also modulated by the activity of cell surface ANT (Biochim. Biophys. Acta 2017, 1862, 832–841). Please update in the text and Figure 4 (left panel). Alternatively, as presented in the right panel of Figure 4, ProTα could directly stimulate the ATPase activity of F1-ATPase at the cell surface to generate extracellular ADP. Please mention this possibility in the text.
Minor comments:
Section 1:
- Page 2 lane 1: Delete “in this mechanism”.
Section 3:
- Page 2: define serum free D/F
- Page 3: Define LOG as “low oxygen glucose”
Section 4:
- Page 4 (4.1): typo error missing i at ischemia.
Author Response
Thank you very much for your kind comments. I revised the manuscript according to the reviewer's comments, as follows;
1) Overlapped description of the method used in the prothymosin alpha discovery in Fig. 2 and 3 was carefully revised according to the reviewer's comments.
2) The validity of Gi-mediated PLC activation was carefully explained by adding detailed references (see page 3, lines 11-16). As most recent study revealed that Gi-derived βγ-mediated PLC activation is dependent on Gaq binding to PLC, we revised the Figure 4 and related description in the text (page 7, lines 11-14). Eventually, I revised the Figure 4 and the text by including the possibility of the involvement of unidentified Gq-coupled P2Y. In this Figure 4, I deleted the E-NTDPase involvement, since there are many candidates.
3) The description of ProTa-binding to F1-ATPase subunits was revised according to the reviewer's comments (page 6, lines last 1 and 2). And unnecessary sentence was deleted.
4) Regarding ProTa-induced ATP production, I added the discussion whether ATPase inhibition is involved by citing the reference the reviewer suggested (Page 7, second paragraph lines 1-6).
5) Regarding ADP production, I revised it by inclusion of the reference of ANT, according to the reviewer's comments (Page 7, second paragraph lines 7-13).
6) Minor points were appropriately revised, according to the reviewer's comments.
Round 2
Reviewer 2 Report
The authors responded satisfactorily to all my comments. The review is acceptable for publication.
One minor comment:
The sentence page 10 “(...)and in the suppression of apolipoprotein A1 in endothelial nitric oxide synthesis and vasorelaxation” is confusing. It is actually an activation promoted by F1-ATPase/P2Y pathway. “(...)and in endothelial nitric oxide synthesis and vasorelaxation mediated by apolipoprotein A-1”